# The Effects of Positive End Expiratory Pressure and Lung Volume on Diaphragm Thickness and Thickening

**DOI:** 10.3390/diagnostics13061157

**Published:** 2023-03-17

**Authors:** Paolo Formenti, Sara Miori, Andrea Galimberti, Michele Umbrello

**Affiliations:** 1SC Anestesia e Rianimazione I, ASST Santi Paolo e Carlo—Polo Universitario, Ospedale San Paolo, 20142 Milan, Italy; 2SC Anestesia e Rianimazione I, Ospedale Santa Chiara, APSS, 30014 Trento, Italy; 3SC Anestesia Rianimazione e Terapia Intensiva ASST Nord Milano Ospedale Bassini, 20092 Cinisello Balsamo, Italy; 4SC Anestesia e Rianimazione II, ASST Santi Paolo e Carlo—Polo Universitario, Ospedale San Carlo Borromeo, 20148 Milan, Italy

**Keywords:** diaphragm, ultrasound, ARDS, lung volume, PEEP

## Abstract

**Introduction**: Diaphragm dysfunction is common in patients undergoing mechanical ventilation. The application of positive end-expiratory pressure (PEEP) and the varying end-expiratory lung volume cause changes in diaphragm geometry. We aimed to assess the impact of the level of PEEP and lung inflation on diaphragm thickness, thickening fraction and displacement. **Methods**: An observational study in a mixed medical and surgical ICU was conducted. The patients underwent a PEEP-titration trial with the application of three random levels of PEEP: 0 cmH_2_O (PEEP0), 8 cmH_2_O (PEEP8) and 15 cmH_2_O (PEEP15). At each step, the indices of respiratory effort were assessed, together with arterial blood and diaphragm ultrasound; end-expiratory lung volume was measured. **Results**: 14 patients were enrolled. The tidal volume, diaphragm displacement and thickening fraction were significantly lower with higher levels of PEEP, while both the expiratory and inspiratory thickness increased with higher PEEP levels. The inspiratory effort, as evaluated by the esophageal pressure swing, was unchanged. Both the diaphragm thickening fraction and displacement were significantly correlated with inspiratory effort in the whole dataset. For both measurements, the correlation was stronger at lower levels of PEEP. The difference in the diaphragm thickening fraction during tidal breathing between PEEP 15 and PEEP 0 was negatively related to the change in the functional residual capacity and the change in alveolar dead space. **Conclusions**: Different levels of PEEP significantly modified the diaphragmatic thickness and thickening fraction, showing a PEEP-induced decrease in the diaphragm contractile efficiency. When using ultrasound to assess diaphragm size and function, the potential effect of lung inflation should be taken into account.

## 1. Introduction

Diaphragm ultrasonography is a rapidly expanding field in pulmonology and intensive care [1]. It is a feasible and accurate tool to assess diaphragm anatomy and physiology. Among the different applications of diaphragm ultrasonography [2], the assessment of end-expiratory thickness and its change over time have been suggested as indices of the progressive atrophy of the diaphragm and the effects of ventilator-induced diaphragm dysfunction [3,4,5]. On the other hand, the diaphragm inspiratory thickening fraction (TF) has been reported to correlate with the inspiratory effort measured by either transdiaphragmatic or esophageal pressure changes, both in healthy and in ventilated individuals [6,7,8,9]. However, despite its extensive use [10,11], conflicting data are emerging concerning the extent and the reliability with which the TF may reflect the inspiratory pressure generated by the diaphragm, that is, the physiologic estimate of the diaphragm function [12]. In fact, very few studies have reported correlation values for the diaphragm TF change in the inspiratory effort relationship [13], and some recent investigations even questioned the strength and clinical significance of the association between diaphragm TF and the generation of inspiratory pressure, as assessed either by the esophageal or the transdiaphragmatic pressure swings [14].

The authors of those studies provided different possible explanations for this lack of correlation. Among them, a nonlinear pressure–volume relationship of the diaphragm, or that the diaphragm TF and the pressure generated depend on the specific pattern of thoraco-abdominal motion. Alternatively, the application of PEEP and the varying end-expiratory lung volume causes changes in diaphragm geometry leading to an altered force-generating capacity of the diaphragm because of a different position of the diaphragm muscle over its force/length relationship. Although interesting, some of the mechanisms proposed above have not been formally tested. The main aim of the present research was to assess the impact of the level of positive end-expiratory pressure on diaphragm thickness, TF and displacement. The secondary aim was the analysis of the relationship between diaphragm TF and inspiratory effort at different levels of PEEP and the investigation of the determinants of such relationship.

## 2. Materials and Methods

### 2.1. Subjects

We conducted a single center, observational study in a mixed medical and surgical ICU in a university-affiliated hospital. The study was approved by the Institutional Review Board (Comitato Etico Interaziendale Milano Area 1, 2015/ST/029), and written informed consent was obtained according to Italian regulations. Consecutive patients aged 18 or older who were spontaneously breathing via tracheostomy with a CPAP of ≤10 cmH_2_O and had previously been invasively ventilated for acute respiratory failure were eligible for inclusion. The exclusion criteria included hemodynamic instability requiring vasopressors, hypoxemia requiring PEEP > 10 cmH_2_O and/or FiO_2_ > 60%, need for inspiratory pressure support, Richmond Agitation and Sedation Scale score < −1, preexisting neurological weakness or a diagnosis of ICU-acquired weakness, diagnosis of COPD, pregnancy and malignancy.

### 2.2. Study Protocol

According to the local weaning protocol for patients recovering from acute respiratory failure and with a tracheostomy, the patients were assisted with continuous positive airway pressure using a standard intensive care mechanical ventilator (Servoi, Maquet, Solna, Sweden). The patients underwent a PEEP-titration trial with the application of three random levels of PEEP, lasting 30 min each, in the semi-recumbent position: 0 cmH_2_O (PEEP0), 8 cmH_2_O (PEEP8) and 15 cmH_2_O (PEEP15); FiO_2_ was unchanged. During the last 5 min of each step, the pattern of breathing and indices of respiratory effort were assessed, arterial blood was sampled for gas determinations and hemodynamic parameters were recorded, as well as diaphragm ultrasound; end-expiratory lung volume was measured. At the end of each step, the patients were instructed to make a maximal inspiration, and the same measurements as in spontaneous tidal breathing were taken. The patients could be lightly sedated to ensure comfort, according to the local clinical practice, and during the study the sedation level was not modified; the level of sedation (RASS scale) and the sedatives used were recorded. The protocol was allowed to be stopped if the patient developed signs of respiratory distress (respiratory rate > 35 breaths/min, SpO_2_ < 90%, heart rate > 140 beats/min, systolic blood pressure > 180 mmHg, diaphoresis or anxiety).

### 2.3. Measurements

Esophageal pressure was measured by an esophageal balloon catheter (Smart Cath, Viasys, Palm Springs, CA, USA) connected to a pressure transducer. The position of the balloon in the esophagus and its filling volume were optimized to obtain a ratio between the esophageal and airway pressure swings during occlusion [15]. The esophageal pressure swing (ΔPes) was calculated as the maximal negative deflection of the esophageal pressure from the end expiratory value, and it was considered as an index of the inspiratory effort [8]. The P0.1 [16] and Pmusc indexes [17] were measured during respiratory pauses. The end-expiratory lung volume (EELV) was measured with a simplified helium dilution technique, as previously described [18]. A calibrated helium analyzer measured the helium concentration in the balloon, and EELV was calculated using the standard formula:(1)$$EELV\;l=Vb*\frac{Ci}{Cf}-Vb$$
where *Ci* is the helium concentration of the known gas mixture, *Cf* is the final helium concentration and *Vb* is the volume of gas in the balloon.

Alveolar dead space (*Vd*/*Vt_alv_*) was measured according to the Enghoff modification of the Bohr equation:(2)$$Vd/Vt_{alv}=\frac{PaCO_{2}-P_{ET}CO_{2}}{PaCO_{2}}*VT$$

All measures were taken in the semi-recumbent position and were performed during a stable spontaneous breathing pattern of 5 min.

### 2.4. Ultrasonographic Measurements

Ultrasonography was performed by using a multiprobe ultrasound (Mindray TE7 Ultrasound System—Mindray Medical International, Shenzhen, China). A well-trained operator (PF) performed the measurements as previously described [7]. The diaphragmatic thicknesses were assessed by B-mode ultrasonography placed above the right 10th rib in the midaxillary line. The TF was calculated as:(3)$$TF=\frac{\left(end-inspiratory\;thickness-end-expiratory\;thickness\;\right)}{end-expiratory\;thickness}\times 100$$

Diaphragm excursion was measured using a phased array probe, with the probe positioned in the subcostal margin in the midclavicular line, with the aim of imaging the posterior third of the diaphragm. Its displacement was then visualized using the M-mode. During the study, careful attention was paid to selecting the same breaths for the diaphragm ultrasound and for the esophageal pressure measurements by starting the registration of the esophageal pressure and diaphragm ultrasound in the same moment and considering the same breaths.

### 2.5. Statistical Analysis

The data were analyzed using Stata 13.0 (StataCorp, College Station, TX, USA). Normality was examined with Shapiro–Francia test. The data are presented as the mean and standard deviation, if normally distributed, or median and interquartile range (IQR) otherwise. The variables of interest were compared between groups with the Student’s *t*-test, Mann–Whitney rank-sum test and Fisher’s exact test. We did not perform a formal sample size calculation since a similar topic has not been addressed in previously available publications. The analysis of variance for repeated measurements was used to analyze variables recorded over the three steps (PEEP0, PEEP8 and PEEP15). The statistical significance of the within-subject factors was corrected with the Greenhouse–Geisser method. The nonparametric variables were analyzed using the Friedman test. Pairwise post hoc multiple comparisons were carried out according to the Tukey method. The correlation between diaphragm TF and ΔPes was investigated at each level of PEEP with linear correlation and on all the data using a linear mixed model for repeated measures to deal with the longitudinal structure of our dataset (patients with repeated measurements over time). In the model, the PEEP level was considered the fixed factor, while subjects were the random factor. The association between variables was expressed as the coefficient of determination for the mixed model (marginal R^2^, that is, the attributable variance due to the fixed-effect portion of the model) and *p*-value. Two-tailed *p*-values < 0.05 were considered for statistical significance. Since no formal sample size calculation was performed, given the difference found in end-expiratory diaphragm thickness between PEEP 15 and 0, we calculated that the post hoc power of our investigation, considering an alpha level of 0.05 and a two-sided test, was 0.76.

## 3. Results

We enrolled 14 patients in the study. All of them concluded the CPAP titration trial without adverse events. Diaphragm ultrasound was feasible in all patients. The baseline characteristics of the case-mix at ICU admission and the main hemodynamic and respiratory parameters at enrolment are reported in Table 1. The patients were enrolled after an average of 10 (IQR 8; 13) days from ICU admission. On the day of enrolment in the study, the mean RASS score was 0 (−1; 0), and the pain verbal numerical rating was 0 (0; 1). No patient was receiving any vasoactive drug. The total length of mechanical ventilation was 11 (9; 21) days, and the ICU and hospital mortality were 28.5 and 35.7%, respectively. Table 2 shows the gas exchange and respiratory mechanics during the three-step PEEP trial. As expected, oxygenation significantly increased with higher PEEP levels, as did the functional residual capacity; carbon dioxide, the arterial to end-tidal CO_2_ difference, as well as alveolar dead space, all increased with an increasing PEEP. The respiratory rate, P0.1 and the negative inspiratory force remained unchanged despite the level of PEEP. Table 3 and Figure 1 show a comparison of the tidal volume, diaphragm ultrasound and inspiratory effort over the three step of the study, both during tidal as well as during maximal breathing. Notably, tidal volume, diaphragm displacement and TF were significantly lower with higher levels of PEEP, while both the expiratory and inspiratory thickness increased with higher PEEP levels; inspiratory effort, as evaluated by the esophageal pressure swing, was unchanged. Both diaphragm TF and displacement were significantly correlated with inspiratory effort in the whole dataset (marginal R^2^ = 0.586, *p* < 0.001 and marginal R^2^ = 0.359, *p* < 0.01, respectively). Figure 2 shows the influence of PEEP on the degree of correlation between both diaphragm TF and displacement during tidal breathing. For both measurements, the correlation was stronger at lower levels of PEEP: R^2^ = 0.871 (*p* < 0.001) at PEEP 0, R^2^ = 0.511 (*p* = 0.004) at PEEP 8 and R^2^ = 0.234 (*p* = 0.066) at PEEP 15 for diaphragm TF; R^2^ = 0.715 (*p* < 0.001) at PEEP 0, R^2^ = 0.162 (*p* = 0.154) at PEEP 8 and R^2^ = 0.051 (*p* = 0.437) at PEEP 15 for diaphragm displacement. The difference in diaphragm TF during tidal breathing between PEEP 15 and PEEP 0 was negatively related to the change in the functional residual capacity (R^2^ = 0.392, *p* = 0.022) and the change in alveolar dead space (R^2^ = 0.310, *p* = 0.048); the change in diaphragm end-expiratory thickness was linearly related to the change in the end-expiratory lung volume (R^2^ = 0.560, *p* = 0.002) and the change in alveolar dead space (R^2^ = 0.423, *p* = 0.012) (Figure 3).

## 4. Discussion

The main findings of the present investigation can be summarized as follows: (1) the increase in end-expiratory airway pressure, at least in the patients recovering from acute respiratory failure, was associated with an increased end-expiratory lung volume, an increased oxygenation and a higher alveolar dead space, together with a lower tidal volume and a similar inspiratory effort; (2) diaphragm thickness significantly increased with increasing levels of PEEP, whereas diaphragm TF and displacement were reduced at higher PEEP levels; (3) the correlation between diaphragm TF or displacement and inspiratory effort, as assessed by the esophageal pressure swing, was stronger at lower levels of PEEP; and (4) the larger the increase in the end-expiratory lung volume or the alveolar dead space after the increase in PEEP, the larger the increase in the expiratory thickness of the diaphragm and the decrease in diaphragm TF.

Mechanical ventilation is frequently associated with potential complications, which depend on the degree of lung injury, the lung properties and the mode of ventilation employed. Both ventilator under- and over-assistance can impact on multiple aspects of diaphragm structure and function. This condition, known as ventilator-induced diaphragm dysfunction (VIDD) [16], is characterized by a loss of diaphragm force-generating capacity specifically related to the use of mechanical ventilation, and it can affect other respiratory muscles beyond the diaphragm [17]. Recent findings suggest that diaphragm dysfunction is frequently involved during weaning failure determining poor prognosis at the time of liberation from MV [18]. PEEP is routinely applied in mechanically ventilated patients [19]. It improves gas exchange and respiratory mechanics by increasing EELV [20]. A prolonged application of PEEP causes diaphragm remodeling, and an acute withdrawal of PEEP during ventilator weaning reduces EELV and stretches the atrophied diaphragm fibers causing a detrimental effect on the force generation [21]. The use of ultrasound is currently widely applied to evaluate diaphragm function during weaning, as it has been shown that both an increase and a decrease in diaphragm thickness are associated with diaphragm activity and ventilation outcome [22]. Indeed, we found in our current investigation that the level of lung inflation and the specific PEEP setting have a significant impact on diaphragm thickness. Since it is reasonable to expect PEEP levels to be progressively reduced during ICU stay in patients recovering from acute respiratory failure, caution should be used when relying on diaphragm ultrasound to assess VIDD, as the evolution of diaphragm thickness and TF may, at least in part, represent the evolution of lung inflation on top of the change caused by the muscle atrophy. Our results showed how diaphragm thickness significantly increases with increasing levels of PEEP, whereas diaphragm TF and displacement are reduced at higher PEEP levels. The possible explanation of the increase in inspiratory and expiratory thickness associated with the increase in PEEP could depend on the caudal descent of the diaphragm dome due to the increased lung volumes induced by PEEP. This somehow mimics the physiological descent of the diaphragm dome with inspiration, which leads the zone of apposition to shorten and, hence, since its total volume does not change, to thicken.

A higher PEEP was associated with less spontaneous breathing effort (reflected by the esophageal pressure) during mechanical ventilation in ARDS [23,24], although its distinct mechanism is currently unknown. A recent study on healthy volunteers similarly showed how a higher PEEP decreased the neuromechanical efficiency of the diaphragm [21]. Indeed, a higher lung volume is known to reduce the diaphragm length, resulting in less force generation from the muscle [25]. In a rabbit ARDS model, the capacity of the force generation of the diaphragm (transdiaphragmatic pressure following bilateral phrenic nerve stimulation) decreased by increasing the PEEP, and the effect was mediated by alteration in the end-expiratory lung volume (as evidenced by CT) [26].

We showed a correlation between diaphragm TF or displacement and inspiratory effort, as assessed by the esophageal pressure swing, which was stronger at lower levels of PEEP and became weak or nonsignificant at higher levels. Diaphragm TF has been proposed as an index of diaphragm function. Goligher et al. [22] showed a link between the changes in transdiaphragmatic pressure and diaphragm TF even if with a large variability. Likewise, a few others [6,7,8] showed how diaphragm TF was correlated with esophageal and diaphragm pressure–time products. Again, a large difference in diaphragm pressure–time product values for a given diaphragm TF value were observed. The interindividual differences in the change in transdiaphragmatic pressure–diaphragm TF and diaphragm pressure-time product–diaphragm TF relationships were probably not accounted [14]. In the present study we showed how the larger the increase is in the end-expiratory lung volume or the alveolar dead space after the increase in PEEP, the larger the increase in the expiratory thickness of the diaphragm, and the decrease in diaphragm TF contraction was suggested as the diaphragm TF increased with lung volumes [27,28]. In fact, the pressure change resulting from the diaphragm contraction is represented by the transdiaphragmatic. Thus, the caudal diaphragm displacement increases gastric pressure [29], while different abdominal conformations would result in different gastric pressure reactions for a given diaphragm force production [30]. However, the force generated by the diaphragm contraction could be nonuniform across the muscle. Thus, diaphragm TF may be more related to the diaphragm electrical activity than the pressure it generates, although only preliminary data are available, and more detailed studies are yet to be performed [31]. Diaphragm TF and transdiaphragmatic pressure were recently compared in 14 healthy and 25 mechanically ventilated patients. At the group level, a significant relationship was found between the two variables in both populations; however, this relationship was found to be only moderate in healthy subjects and weak in mechanically ventilated patients, and this prevented the possibility of inferring pressure output from ultrasound recordings so that the authors cautioned against the use of this ultrasound index to assess diaphragm function [14].

An interesting while apparently contradictory finding of our investigation is that a similar tidal swing in esophageal pressure was generated despite different degrees of diaphragm TF at different levels of PEEP. Indeed, it is the contraction of the whole diaphragm that should promote the inspiratory effort, and this includes both the zone of apposition and the crural area. The latter, however, is not generally recorded in diaphragmatic ultrasound. A two-dimensional strain ultrasound speckle tracking study demonstrated that, at least during tidal breathing, the crural area had a negative strain, which is a sign of active shortening of that given segment relative to its length. This likely means that the crural area participates in the development of inspiratory effort at least as the apposition zone during spontaneous breathing [32]. Thus, it cannot be excluded that changes in the shape of the diaphragm after PEEP application may modify the regional diaphragm contractility, leading to a greater participation of the crural area and reduced involvement of the zone of apposition while leading to the development of the same inspiratory effort. Our study has a number of limitations. First, we included a small population, which might restrain the study power. Second, we only assessed the right hemidiaphragm, because the thickness of the right hemidiaphragm is generally feasible and reproducible in the zone of apposition in mechanically ventilated patients. However, in subjects with bilateral measurements, right and left hemidiaphragm thickness at end expiration and diaphragmatic TF were shown to be similar [3]. This limitation is common to other studies on the ultrasonographic assessment of diaphragmatic contractile activity. Since the patients enrolled were in a late, resolving phase, caution should be applied before extending the findings to cases of more severe or acute respiratory failure. Although the effect on diaphragm thickness of the increase in lung volume should, on a theoretical ground, be similar, the more severe the respiratory failure, the stiffer should the lung compliance be and, hence, a lower lung volume variation could be expected given a lower transmission of airway pressures to the chest wall. Therefore, the effects on diaphragmatic performances could be lesser in the case of a more severe form of respiratory failure. Lastly, we did not record data on the previous use of corticosteroids or neuromuscular blocking agents, which are frequently used in patients with moderate/severe ARDS [33]. Indeed, the e use of such drugs is known to potentially affect diaphragm function [34] and future studies should consider the impact of these drugs.

## 5. Conclusions

This study showed the impact of the level of positive end-expiratory pressure and lung inflation on diaphragm anatomy and physiology. Both diaphragm thickness and TF seem to be related to lung volume. Different levels of PEEP significantly modified the diaphragmatic TF, showing a PEEP-induced decrease in the diaphragm contractile efficiency. Until larger studies are performed, we advise caution when using ultrasound to assess diaphragm size and function and suggest taking into account the effect of lung inflation.

## Figures and Tables

**Figure 1 diagnostics-13-01157-f001:**
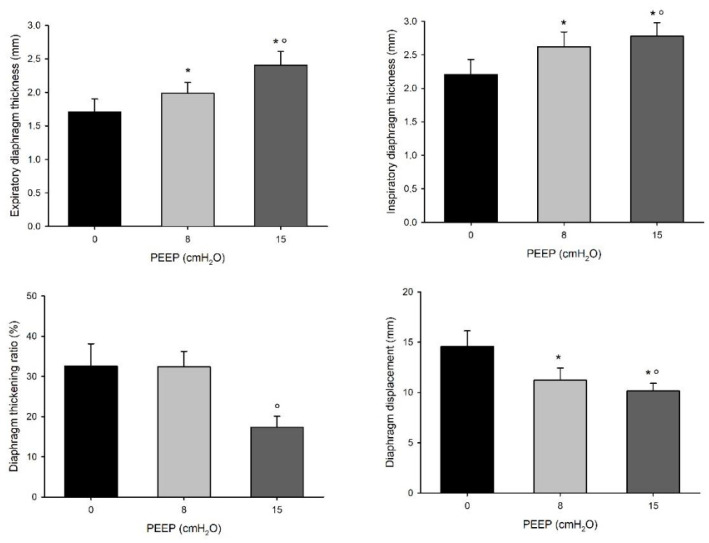
Diaphragm expiratory and inspiratory thickness (upper panel) and TF and displacement (lower panel) during tidal breathing in the three steps of the study. * *p* < 0.05 vs. PEEP0; ° *p* < 0.005 vs. PEEP8.

**Figure 2 diagnostics-13-01157-f002:**
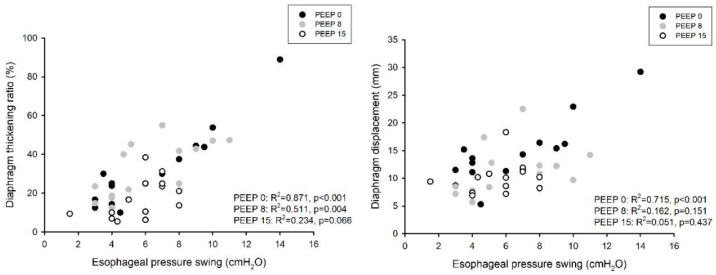
Correlation between inspiratory effort (as evaluated by the inspiratory esophageal pressure swing) and diaphragm TF (**left panel**) and displacement (**right panel**) during tidal breathing in the three steps of the study.

**Figure 3 diagnostics-13-01157-f003:**
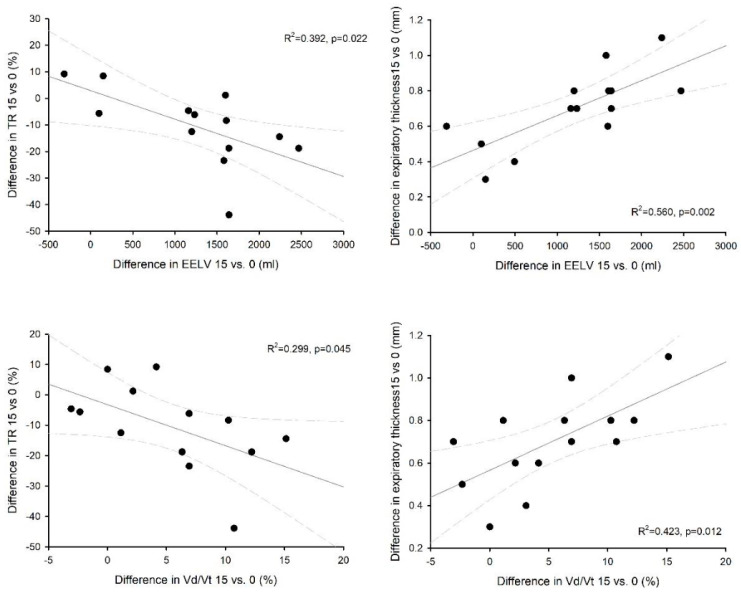
Correlation between the change in diaphragm thickening fraction (TF—left panels) or the change in diaphragm end-expiratory thickness (EELV—right panels) during tidal breathing and the change in the end-expiratory lung volume (upper panels) or in the alveolar dead space (Vd/Vt—lower panels) from PEEP 15 to PEEP 0.

**Table 1 diagnostics-13-01157-t001:** Baseline characteristics of the study population.

Age (years)	77 (63; 78)
Male sex, N (%)	9 (64)
Height (cm)	170.1 ± 5.7
Actual body weight (kg)	76.0 ± 18.9
Ideal body weight (kg)	63.9 ± 5.7
Body mass index (kg/m^2^)	25.7 ± 6.4
Diagnosis:	
Pneumonia	9 (64.4%)
Peritonitis	3 (21.4%)
Pancreatitis	1 (7.1%)
Trauma	1 (7.1%)
SOFA score at ICU admission	7 (4; 8)
SAPS II score at ICU admission	39 (33; 54)
Hemoglobin concentration (g/dL)	10.9 ± 0.9
Heart rate (b/min)	87 ± 11
Mean arterial blood pressure (mmHg)	77.5 ± 7.1
FiO_2_	0.4 (0.3; 0.5)
PEEP (cmH_2_O)	7 (6; 8)
Pressure support (cmH_2_O)	5 (4; 6)
Peripheral oxygen saturation (%)	97.9 ± 1.3
Respiratory rate (1/min)	19.7 ± 4.0

SOFA: Sequential Organ Failure Assessment Score; SAPS II: Simplified Acute Physiology Score, 2nd version; ICU: intensive care unit; FiO_2_: fraction of inspired oxygen; PEEP: positive end-expiratory pressure.

**Table 2 diagnostics-13-01157-t002:** Gas exchange and respiratory mechanics during the three steps of the study.

	PEEP 0	PEEP 8	PEEP 15	*p*
Respiratory rate (min^−1^)	22.1 ± 4.8	23.1 ± 4.8	22.4 ± 4.0	0.5699
pH	7.47 ± 0.04	7.46 ± 0.05	7.45 ± 0.03	0.1384
PaO_2_ (mmHg)	88.5 ± 10.6	110.1 ± 19.7 *	123.3 ± 36.7 *°	0.0030
PaCO_2_ (mmHg)	42.0 ± 4.6	43.3 ± 4.2 *	45.2 ± 4.3 *°	<0.0001
EtCO_2_ (mmHg)	38.3 ± 4.7	39.2 ± 3.1	38.7 ± 3.3	0.3850
Δ A-EtCO_2_ (mmHg)	3.7 ± 1.6	4.1 ± 2.5 *	6.5 ± 1.9 *°	0.0010
Vd/Vt alv (%)	9.0 ± 3.6	9.2 ± 5.1	14.3 ± 3.3 *°	0.0026
EELV (mL)	1066 (660; 2400)	1537 (1300; 3240) *	3094 (2111; 3500) *°	<0.0001
P0.1 (cmH_2_O)	3.1 ± 1.6	2.36 ± 1.27	3.35 ± 1.97	0.0596
NIF (cmH_2_O)	26.7 ± 9.3	31.1 ± 11.4	27.6 ± 8.0	0.7775

PaO_2_: partial pressure of oxygen in the arterial blood; PaCO_2_: partial pressure of carbon dioxide in the arterial blood; ETCO_2_: end-tidal carbon dioxide; Δ A-EtCO_2_: difference between arterial and end-tidal carbon dioxide; EELV: end-expiratory lung volume; P0.1: airway pressure drop in the first 100 ms of an occluded inspiration; NIF: negative inspiratory force. * *p* < 0.05 vs. PEEP 0; ° *p* < 0.05 vs. PEEP 8.

**Table 3 diagnostics-13-01157-t003:** Tidal volume, diaphragm ultrasound and inspiratory effort under tidal and maximal breathing during the three steps of the study.

Tidal Breathing	PEEP 0	PEEP 8	PEEP 15	*p*
Tidal volume (mL)	384 ± 79	367 ± 96	302 ± 81 *°	0.0073
Displacement (mm)	14.6 ± 5.8	11.2 ± 4.5 *	10.1 ± 2.8 *	0.0378
Thickness EI (mm)	2.21 ± 0.83	2.62 ± 0.82 *	2.77 ± 0.75 *°	0.0010
Thickness EE (mm)	1.70 ± 0.72	1.98 ± 0.60 *	2.41 ± 0.75 *°	0.0001
Thickening fraction (%)	32.6 ± 20.8	32.4 ± 14.4	17.4 ± 10.2 *	0.0152
ΔPes (cmH_2_O)	6.4 ± 3.3	6.1 ± 2.7	5.7 ± 1.8	0.6281
**Maximal breathing**	**PEEP 0**	**PEEP 8**	**PEEP 15**	** *p* **
Tidal volume (ml)	905 ± 278 ^#^	813 ± 227 ^#^	667 ± 202 *^#^	0.0240
Displacement (mm)	20.4 ± 12.9	28.9 ± 19.6 ^#^	23.3 ± 14.8 ^#^	0.0795
Thickness EI (mm)	2.63 ± 0.94 ^#^	3.01 ± 1.09 *^#^	3.48 ± 1.23 *°^#^	0.0152
Thickness EE (mm)	1.79 ± 0.79	2.01 ± 0.57 *	2.42 ± 0.58 *°	0.0010
Thickening fraction (%)	53.8 ± 28.4 ^#^	63.5 ± 26.1 ^#^	42.2 ± 28.2 ^#^	0.2684
ΔPes (cmH_2_O)	10.9 ± 4.8 ^#^	11.2 ± 5.3 ^#^	11.4 ± 4.3 ^#^	0.9538

EI: end-inspiratory; EE: end-expiratory; ΔPes: esophageal pressure swing. * *p* < 0.05 vs. PEEP 0; ° *p* < 0.05 vs. PEEP 8. ^#^
*p* < 0.05 vs. tidal breathing.

## Data Availability

The data presented in this study are available on request from the corresponding author. The data are not publicly available due to privacy and ethical restrictions.

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
