# Peer review of "The Effects of Positive End Expiratory Pressure and Lung Volume on Diaphragm Thickness and Thickening"

_diagnostics, 2023, doi:10.3390/diagnostics13061157_

Round 1

Reviewer 1 Report

Some comments:

Page 4, Part 2.5. paragraph 1, line 3. you mention other physiological studies, what are they?

Page 4. part 3.paragraph 1, line 8. Are the references [9;21] in place within the text?

Table 1. Did any of these patients receive corticosteroids? or neuromuscular blocker? during the initial period of ICU.

Author Response

Page 4, Part 2.5. paragraph 1, line 3. you mention other physiological studies, what are they?

We thank the reviewer for his/her observation. We agree that this sentence was misleading, and we modified the text deleting this part.

Page 4. part 3.paragraph 1, line 8. Are the references [9;21] in place within the text?

We thank the reviewer for his/her observation; indeed, the numbers in square brackets are not referred to references, but to the median and interquartile range values. For the sake of clarity, we added [IQR] at the first appearance in the results

Table 1. Did any of these patients receive corticosteroids? or neuromuscular blocker? during the initial period of ICU.

We thank the reviewer for his/her observation. Unfortunately, these interesting data have not been recorded. However, we agree that previous use of corticosteroids or neuromuscular blocking agents can indeed impact the diaphragm function. We added a little discussion of this issue in the limitation section.

Reviewer 2 Report

I have read with much interest the manuscript titled “The Effects of Positive end Expiratory Pressure and Lung Volume on Diaphragm Thickness and Thickening by Dr. Formenti and coworkers.”

In this pilot study, the authors have analyzed the effect of different levels of PEEP in diaphragm ultrasonographic measurements in adult patients with a tracheostomy and CPAP <=10 cmH20 after ARDS. They have mainly demonstrated that changes in PEEP are related to their principial outcomes: increase in PEEP was correlated with an increase in lung volume, diaphragm displacement and thickness, with the same work of breathing (measured by esophageal pressure).

These results are very interesting, as they have not been described before, and they reinforce the need of taking into account the amount of respiratory support when evaluating diaphragm ultrasound in critical care patients. 

I will only suggest some minor corrections/suggestions:

1.     As sample size cannot be calculated, at least the power of the sample should be provided.

2.     Tables should include the number of patients in the whole sample (n=14). Table 1 and 2 could be merged.

3.     Were the values of diaphragm ultrasound and inspiratory effort (table 4) compared between the tidal and maximal respirations? I think it would be interesting.

4.     The authors use indistinctly the terms “thickening” and “expiratory and inspiratory thickening” to express opposite variables, although only normal diaphragm thickness was described in the methods section. In page 4: “Notably, tidal volume, diaphragm displacement and thickening were significantly lower with higher levels of PEEP, while both expiratory and inspiratory thick-ness increased with higher PEEP levels”. This is repeated in the manuscript, and I don’t understand the differences between the two concepts: maybe does “thickening” stand for “thickening ratio”? It should be explained.

5.     The second paragraph of the discussion section is a review of concepts that should be included in the introduction, if needed, not in the discussion section.

6.     I would like to read a possible explanation of the increase in inspiratory and espiratory thickness with increase of PEEP.

7.     The population studied are patients with low respiratory support (PEEP 7 cmH20 and PS 5 cmH20): would the results of this study different in patients with a more severe or acute disease?

8.     References:

-       Number 2 is incomplete: it lacks authors’ names, journal’s name, year, number and pages.

-       An interesting reference on the effect of PEEP increasing/decreasing in ARDS and its relationship with diaphragm excursion is lacking: Cho et al. Resp Care 2021.

-       Refs. 24 and 28 are the same.

Author Response

I will only suggest some minor corrections/suggestions:

1.As sample size cannot be calculated, at least the power of the sample should be provided.

We thank the reviewer for this interesting issue which was missing in our manuscript. The text has been amended and now reads: “Since no formal sample size calculation was performed, given the difference found in end-expiratory diaphragm thickness between PEEP 15 and 0, we calculated that the post-hoc power of our investigation, considering an alpha level of 0.05 and a two-sided test, was 0.76”.

2. Tables should include the number of patients in the whole sample (n=14). Table 1 and 2 could be merged.

We thank the reviewer for these comments. We merged tables 1 and 2 and modified the number of patients as suggested.

3.Were the values of diaphragm ultrasound and inspiratory effort (table 4) compared between the tidal and maximal respirations? I think it would be interesting.

We thank the reviewer for his/her observation. We compared the tidal volume, diaphragm ultrasound and inspiratory effort between tidal and maximal breathing during the three steps of the study; results are now reported in the table.

4.the authors use indistinctly the terms “thickening” and “expiratory and inspiratory thickening” to express opposite variables, although only normal diaphragm thickness was described in the methods section. In page 4: “Notably, tidal volume, diaphragm displacement and thickening were significantly lower with higher levels of PEEP, while both expiratory and inspiratory thick-ness increased with higher PEEP levels”. This is repeated in the manuscript, and I don’t understand the differences between the two concepts: maybe does “thickening” stand for “thickening ratio”? It should be explained.

We thank the reviewers for these observations. The term thickening always referred to thickening fraction, but we agree that it was potentially misleading. We modified all over the text to better clarify this point.

5. The second paragraph of the discussion section is a review of concepts that should be included in the introduction, if needed, not in the discussion section.

We thank the reviewer for this suggestion. We modified the paragraph as suggested.

6. I would like to read a possible explanation of the increase in inspiratory and espiratory thickness with increase of PEEP.

We thank the reviewer for this suggestion. The possible explanation of the increase in inspiratory and expiratory thickness with increase of PEEP could be due to the caudal descent of the diaphragm dome due to increased lung volume induced by PEEP. This should somehow mimic the physiological descent of the diaphragm dome with inspiration, that leads the zone of apposition to shorten and hence, since its total volume does not change, to thicken. We added this part in the discussion.

7. The population studied are patients with low respiratory support (PEEP 7 cmH20 and PS 5 cmH20): would the results of this study different in patients with a more severe or acute disease?

We thank the reviewer for this observation. Unfortunately, we cannot answer this question. We could only speculate that the grater the respiratory failure is, the stiffer should the lung compliance be, and hence and hence a lower lung volume variation could be expected given a lower transmission of airway pressures to the chest wall after application of PEEP. Therefore, the effects on diaphragmatic performances could be lesser in case of a more severe form of respiratory failure. A brief sentence was added in the discussion.

8. References:

- Number 2 is incomplete: it lacks authors’ names, journal’s name, year, number and pages.

We modified the reference.

- An interesting reference on the effect of PEEP increasing/decreasing in ARDS and its relationship with diaphragm excursion is lacking: Cho et al. Resp Care 2021.

We added the paper suggested as reference #36

- Refs. 24 and 28 are the same.

We checked the references and deleted duplicates. Thanks.